# Toward Timeliness-Enhanced Loss Recovery for Large-Scale Live Streaming

## ABSTRACT

Due to the limited permissions for upgrading dual-side (i.e., server-side and client-side) loss tolerance schemes from the perspective of CDN vendors in a multi-supplier market, modern large-scale live streaming services are still using the automatic-repeat-request (ARQ) based paradigm for loss recovery, which only requires server-side modifications. In this paper, we first conduct a large-scale measurement study with a collection of up to 50 million live streams. We find that loss shows *dynamics* and live streaming contains frequent *on-off mode switching* in the wild. We further find that the recovery latency, enlarged by the ubiquitous retransmission loss, is a critical factor affecting client-side QoE (e.g., video freezing) of live streaming. We then propose an enhanced recovery mechanism called AutoRec, which can transform the disadvantages of on-off mode switching into an advantage for reducing loss recovery latency without any modifications on the client side. AutoRec also adopts an online learning-based policy to fit the dynamics of loss, balancing the tradeoff between the recovery latency and the incurred overhead. We implement AutoRec upon QUIC and evaluate it via both testbed and real-world deployments of commercial services. The experimental results demonstrate the practicability and profitability of AutoRec, in which the 95th-percentile times and duration of client-side video freezing can be lowered by 34.1% and 16.0%, respectively.

## 1 INTRODUCTION

Internet live services such as Youtube Live, TikTok Live, and Twitch have gradually become a fundamental element for enriching daily life and work [1, 2]. This also introduces an urgent requirement to promote the transmission performance of live streaming. The ubiquitous packet loss is an essential factor affecting client-side quality-of-experience (QoE) [3, 4], which will introduce head-of-line (HOL) blocking and even incur long-time video freezing if the available frames in the player buffer are all consumed. Therefore, loss tolerance control matters in live-streaming services.

The existing loss tolerance schemes mainly focus on designing dual-side (i.e., service-side CDN and client-side application) control policies, including Forward Error Correction (FEC) [5–7], multi-path retransmissions [8, 9], semi-reliable transmissions [10, 11], and application-level controls [12–14]. However, as shown in Figure 1, live-streaming application operators (e.g., TikTok Live) usually apply the Multi-Supplier Strategy [15] in the CDN market. As a result,

*ACM Multimedia 2024, 28 October - 1 November 2024, Melbourne, Australia*

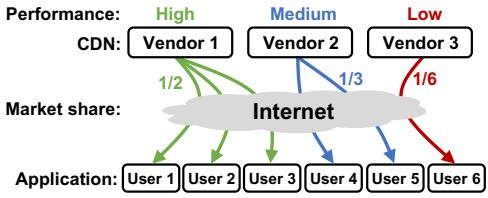

**Figure 1: An example of a multi-supplier market for CDN vendors in large-scale live streaming services.**

it is the CDN vendor's duty that optimize the transmission performance (e.g., loss tolerance), according to which the application operators will choose the better-performed CDN vendors to carry more traffic (i.e., larger market share). In this case, only server-side sending policies can be adjusted by the selected CDN vendors, which lack the proper authority to synchronize client-side control rules. Thus, the above-mentioned arts for loss tolerance control suffer from deployment issues in the multi-supplier CDN market.

In this case, most modern CDN vendors only apply the automatic-repeat-request (ARQ) paradigm [16] to control loss tolerance as the commercial solution, which retransmits only one replica of the lost packet when a loss is detected. However, we find that the legacy ARQ-based loss recovery is far from satisfactory according to our performed large-scale measurements. For example, the proportion of connections with maximum retransmission times of two or more exceeds 43%. Among them, a considerable portion of the connections has certain packets that are retransmitted even more than 10 times (§2.1). The retransmission loss enlarges the loss recovery latency by 123.2ms on average and 279.3ms in the worst case. This enlarged recovery latency further increases the probability of empty buffer space on the client side, thereby increasing the risk of video freezing (§2.3). Thus from a philosophical standpoint, it is worth asking: Can we accelerate loss recovery solely through CDN servers without modifying clients? How can it be addressed without adding significant additional overhead?

Our key insight is that the on-off mode switching ubiquitously occurs in current live streams (§2.2), where the bandwidth during the "off" periods is not fully utilized. Based on our measurements, it can be observed that each live stream spends 464 ms in off-mode per second and enters off-mode 20 times per second on average. It is well-studied that the on-off traffic pattern is not conducive to transmission control [17–22]. However, we argue that it can act as an advantage for the loss tolerance control of live streaming. In this paper, we present AutoRec, an enhanced loss recovery mechanism that can transform the disadvantages of on-off mode into an advantage of loss tolerance controls under large-scale live streaming.

A straightforward way of AutoRec is to directly reinject a fixed number of replicas of loss packets once stepping into off-streaming mode. However, this approach faces two challenges. **First**, the fixed settings of redundant replicas cannot well adapt to the dynamics of packet loss, while excessive reinjection of packets results in

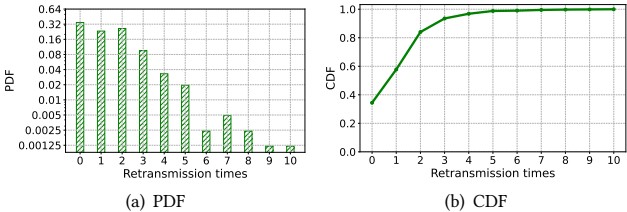

(a) PDF        (b) CDF

**Figure 2: The maximum retransmission times in the wild.**

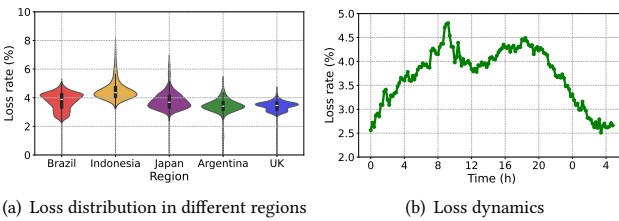

(a) Loss distribution in different regions    (b) Loss dynamics

**Figure 3: Examples of loss dynamics in the wild.**

non-trivial recovery overhead. **Second**, according to our large-scale measurement study, the off-mode is unevenly distributed throughout an entire live session, the sending of replicas might be delayed when the stream doesn't enter off-mode in time. To tackle these issues, we propose a two-step solution: Redundancy Adaption and Reinjection Control.

*Redundancy Adaption* smartly determines how many replicas of the lost packets should be reinjected. Specifically, it adopts an *online learning-based policy* to dynamically set the number of replicas. The goal is to send the least number of replicas that adapt to the dynamics of packet loss. This assures a minimized redundancy overhead while accelerating loss recovery.

*Reinjection Control* determines when to retransmit the replicas. Generally, it enables the replica reinjection during the off-modes. AutoRec further adopts the *opportunistic reinjection* to trigger loss reinjection even though lacking the desired opportunity of off-mode. This assures that each replica can be reinjected in time even under uneven distribution of off-modes.

We implement the AutoRec prototype in the user-space QUIC protocol and deploy it on both testbed and real-network CDN proxy for 6 months. The experimental results demonstrate the practicability and profitability of AutoRec, in which the average (95th-percentile) times and duration of client-side video freezing can be lowered by 11.4% and 5.2% (34.1% and 16.0%), respectively.

The rest of the paper is organized as follows: §2 motivates our work with a large-scale measurement study. Then, the high-level architecture and design details of AutoRec are depicted in §3 and §4, respectively. §5 gives the experimental evaluations of AutoRec. §6 overviews the related work and §7 concludes the paper.

## 2 MEASUREMENT STUDY

In this section, we conduct a large-scale measurement study to motivate our work. We first explore the characteristics of loss (§2.1) and the characteristics of live streaming (§2.2) in the wild. We then analyze the performance of live streaming under the context of packet loss, especially under retransmission loss (§2.3).

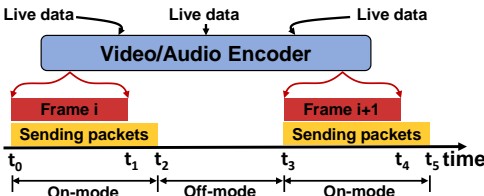

**Figure 4: The sketch of on-off mode in live streams.**

### 2.1 Characteristics of Loss

We dissected the characteristics of loss using real-world logs spanning up to two weeks. These logs documented specific details related to packet loss, including the frequency of loss events and the magnitude of each loss. Our measurements encompass over 200,000 connections, spanning a variety of application scenarios and regions worldwide. The findings are as follows.

**Observation #1: Retransmission loss is ubiquitous.** A packet may undergo multiple retransmissions before the receiver correctly receives it. We can delve deeper into the number of times each packet was retransmitted before successful reception. For any given connection, the maximum retransmission times can be calculated as the highest count of retransmissions across all packets within that connection. Figure 2 illustrates the distribution of the maximum retransmission times in the production network. We discover that the proportion of connections with maximum retransmission times of two or more exceeds 43%. Among them, a considerable portion of the connections have certain packets that are retransmitted even more than 10 times. Traditional ARQ mechanism focuses solely on promptly retransmitting lost packets after each loss event, but it overlooks the total number of retransmissions and the total time required for the retransmitted data to be successfully received by the receiver. Therefore, the traditional ARQ mechanism hardly meets the timeliness requirements for data in certain scenarios (e.g., live streaming).

**Observation #2: Loss shows dynamics.** We then investigate the distribution of the loss rate (every 5 minutes) for each connection across various global regions. The results are depicted in Figure 3(a). While some regions exhibit similar average packet loss rates, their packet loss rate deviations differ (i.e., the violin shapes are different). For instance, both Brazil and Japan have an average packet loss rate of 3.78%, but Japan has the highest packet loss rate of 7.1% while Argentina has only 5.7%. Figure 3(b) further illustrates how the packet loss rate evolves. Specifically, the loss rate is never static, varying between 2% and 5% over 24 hours. This confirms that packet loss exhibits dynamic behavior in real-world scenarios. This also reveals that an adaptive loss tolerance scheme should adapt to differentiated conditions and the always-changed status of networks.

### 2.2 Characteristics of Live Streaming

Unlike video-on-demand or file traffic, the current live streams frequently and extensively reveal off-mode, in which a sender temporarily has no data (i.e., becomes application-limited [19][23]) for continuous transmissions after sending one or more frames. On the one hand, the new generation rate of live data might be slower (e.g., than the sending rate) so senders have to wait for a while until

**Table 1: Live-streaming recovery latency measurement results.**

| metric | value | loss rate | | | | SRTT(ms) | | | |
|---|---|---|---|---|---|---|---|---|---|
| | | 0% ~ 3% | 3% ~ 10% | 10% ~ 30% | 30% ~ 50% | 0 ~ 50 | 50 ~ 200 | 200 ~ 500 | 500 ~ 2000 |
| recovery latency (ms)* | 123.2 | 26.8 | 94.5 | 186.4 | 530.2 | 68.2 | 170.2 | 273.6 | 418.0 |
| maximum recovery latency (ms) | 279.3 | 51.8 | 204.9 | 451.3 | 1288.4 | 149.2 | 391.3 | 698.8 | 960.9 |

*  The displayed recovery latency (maximum recovery latency) only records the each-stream's average recovery latency (maximum recovery latency) of the lost data, whose recovery requires two or more retransmissions.

their sending queues contain data again. For CDN vendors, the data generation rate reflects the traffic transmission rate from anchors to servers, which can be easily affected by real-time network status. On the other hand, the live data will be encoded into video or audio frames based on selected frame rate(s) and bitrate before its transmission, which can also introduce time intervals between adjacent frames. If one frame has been sent out while the follow-up frame has not yet been encoded, CDN senders have to enter off-mode.

As Figure 4 shows, the data of live streaming is encoded (by video/audio encoder) into frame i which is delivered to the sending queue (from $t_0$ to $t_1$) and is sent out before $t_2$. In this case, the sender has to keep waiting (from $t_2$ to $t_3$) for frame i+1 which will be encoded based on the follow-up live data. In this paper, the on-off mode that appears on the sender side can be recognized with the following conditions.

- **On-mode:** A mode that occurs when enough data exists in the sending queue (i.e., $t_0 \sim t_2$ and $t_3 \sim t_5$ in Figure 4), which can be sent by a sender at the subsequent time.
- **Off-mode:** A mode that occurs when no data can be obtained for traffic transmissions (i.e., $t_2 \sim t_3$ in Figure 4), making senders have to wait for the follow-up frame.

We make large-scale measurements and gain the following observations to explore the characteristics of on-off modes in live streaming.

**Observation #1: The on-off mode switching commonly exists in live streaming.** Figure 5(a) presents our measurements of the duration each stream spends in off-mode every second, as well as the frequency of each stream entering off-mode on a per-second basis, where we can learn more than 95% of the live streams can be in off-mode for 394ms per second and enter off-mode 17 times per second. On average, each live stream spends 464ms in off-mode per second and enters off-mode 20 times per second.

**Observation #2: The off-modes are unevenly distributed throughout the live-streaming lifetime.** Figure 5(b) depicts the cumulative values of both off-mode duration and detected loss amount in our testbed experiments. We can find (i) off-mode mainly occurs after 2s while only 2 out of 7 packets are detected lost during this period; (ii) more (i.e., 5 out of 7) packet losses "meet" less off-mode within the first second of this measured stream, in which, especially, this streaming is always keeping on-mode traffic transmission when the 4-th to 6-th losses are detected.

## 2.3 Live Streaming Performance under Loss

Live streaming has a high requirement for data timeliness that can affect and reflect client-side QoE, in which the traffic data that fails to reach receivers in time will cause longer intra-stream HOL

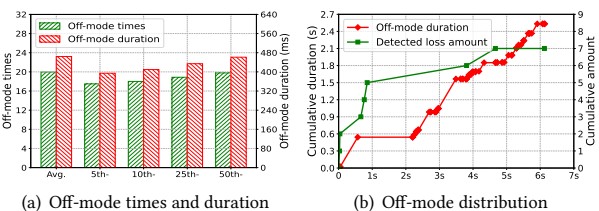

(a) Off-mode times and duration    (b) Off-mode distribution

**Figure 5: On-off mode measurements and experiments.**

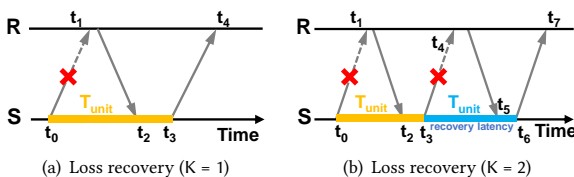

(a) Loss recovery (K = 1)    (b) Loss recovery (K = 2)

**Figure 6: Loss recovery between sender(S) and receiver(R).**

blocking time, especially in the common designs of TCP and QUIC protocol [24], introducing video freezes of players, e.g., Tiktok Live and Youtube Live. Due to the characteristics of packet loss and the limitations of traditional ARQ (§2.1), the timeliness issues caused by packet loss in live streaming deserve attention. We introduce recovery latency to measure the timeliness of packet loss recovery and conduct large-scale measurements on live streams in real networks.

*Recovery latency* is defined as the duration from when any data is detected lost to when resending a recovery packet *that will be successfully received.* $T_{unit}$ is defined as the sum of the delayed time of ACK packets, RTT, and loss detection time, representing the time elapsed for a data packet from being sent to being retransmitted. recovery latency consists of zero or more $T_{unit}$, reflecting the additional time for loss recovery besides first $T_{unit}$. In Figure 6(a), recovery latency is close to 0, in which both the sender-side loss identification and retransmission all occur at $t_3$. Figure 6(b) shows the resent data is detected lost again and another recovery packet (*that will be successfully received* at $t_7$) is sent at $t_6$, where recovery latency = $t_6 - t_3$. Besides, the maximum recovery latency is employed to evaluate the largest recovery latency for loss recoveries.

We make large-scale measurements and collect the transmission logs of 50 million live streams. We then classify the average values of measured recovery latency and maximum recovery latency based on the ranges of loss rate and smooth RTT (SRTT), as Table 1 shows.

**Observation #1: The current recovery latency of live streaming is far from satisfactory.** For the lost data that requires more than 1-time retransmissions, traffic senders waste 123.2 ms (i.e.,

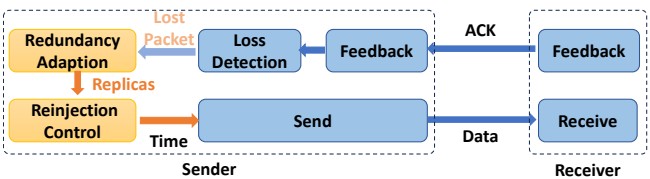

**Figure 7: The architecture of AutoRec.**

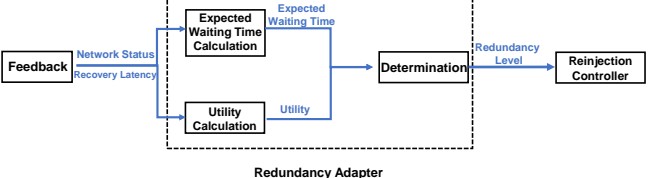

**Figure 8: Redundancy Adapter.**

recovery latency=123.2 ms), on average, before sending out the recovery packets that their receivers will indeed acknowledge. Worse still, the average maximum recovery latency is up to 279.3 ms, which can easily cause video freezing.

**Observation #2: Higher loss rate and larger SRTT can easily introduce more deteriorated recovery latency.** As Table 1 shows, with the increase in loss rate, the recovery latency is also gradually increasing. This is because the recovery data can easily suffer from another packet loss in the networks with higher loss rate. By contrast, SRTT has serious impacts on recovery latency. For example, maximum recovery latency will be improved to ~1 s, on average, under the SRTT of 500 ~ 2000 ms, which is unbearable for the timeliness of loss recovery.

Therefore, an enhanced recovery scheme is highly required to further optimize the current unsatisfied recovery latency and promote the recovery timeliness of live streams.

## 3 THE AUTOREC OVERVIEW

In this section, we first discuss the design principles of AutoRec. Then we give an overview of the architecture of AutoRec.

### 3.1 Design principles

AutoRec regards the control-unfriendly on-off mode as an essential opportunity for enhancing the recovery latency of live streaming and mitigating the negative effects on player freezing caused by potential HOL blocking. AutoRec should follow two design principles for optimizing the timeliness of loss recovery in live streams.

**Principle #1: Maximize the utilization of specific off-mode to enhance recovery latency.** Concretely, more off-mode that is unfriendly to transmission controls can be leveraged to reinject loss duplicates and further lower recovery latency of live streaming.

**Principle #2: Minimize the non-trivial effects on the follow-up traffic transmissions.** Concretely, the off-mode reinjection cannot consume much more sender-side or network resources so that the transmission efficiency will not seriously deteriorate.

### 3.2 The Architecture of AutoRec

AutoRec is a sender-side modification to the protocol stack whose key modules are illustrated in Figure 7. Particularly, once a loss is detected, AutoRec adopts redundancy adaption to compute the number of replicas of the lost packet that should be retransmitted next. Given the number of replicas, AutoRec then adopts reinjection control to determine the specific order and time for sending out each replica from the sender.

**Redundancy adaptation.** This module answers the question of how many replicas should be sent to accelerate loss recovery. The

redundancy level is defined as the number of replicas that should be resent for a specific lost packet. We define the *reinjection overhead* as the total number of replicas that are sent during transmission. To adapt to the dynamics of loss, we incorporate redundancy adaptation to allow the redundancy level to vary dynamically. To achieve this, we introduce the *Redundancy Adapter*, which gradually learns the feature of loss dynamics and carefully selects the most appropriate redundancy level for each retransmission round of lost packets. This ensures that AutoRec can adapt to the dynamics of packet loss while minimizing the redundancy cost.

**Reinjection control.** This module answers the question of how to send the given number of replicas determined by the Redundancy Adapter. For each lost packet, more than one replica might be injected into the network. To avoid bandwidth contention for unlost data transmission, we introduce the *Reinjection Controller* to enable sending replicas of the lost packet if the stream is in off-mode. To further reduce the loss recovery latency, the Reinjection Controller also opportunistically captures the ideal chance to reinject replicas even when the stream is not in the off-mode. The Reinjection Controller can also keep the bandwidth contention for unlost data transmission within a safe limit. This ensures that AutoRec can adapt to the uneven distribution of off-modes.

## 4 DETAILED DESIGN

In this section, we give the detailed design of AutoRec for its practical deployment.

### 4.1 Redundancy Adapter

To better balance the tradeoff between the targeted recovery latency and the potential reinjection overhead, we introduce the Redundancy Adapter to applies an online learning-based scheme to adjust the redundancy level (denoted by $A_{thres}$) for each loss recovery dynamically. The goal is to reinject "few but enough" replicas for optimizing data timeliness affected by packet losses. Specifically, as illustrated in Figure 8, for each decision interval (denoted by DI), AutoRec performs the following three steps. **Step #1:** Evaluate the utility and its change ratio. **Step #2:** Estimate the expected waiting time at the receiver. **Step #3:** Determine the redundancy level.

**Calculation of the utility.** The utility function $U(A_{thres})$ can be evaluated using the incurred recovery latency and the obtained network status, e.g., minimum RTT (minRTT) and goodput, as Eq. 1 shows. The decreased u shows receivers can take shorter time for loss recoveries without significantly lowering goodput or incurring cost-sensitive $A_{thres}$. Then the utility can be evaluated as follows:

$$U(A_{thres}) = \frac{\text{recovery latency} \cdot \text{sigmoid}_\alpha(A_{thres} - 2)}{\text{minRTT} \cdot \text{goodput}} \quad (1)$$

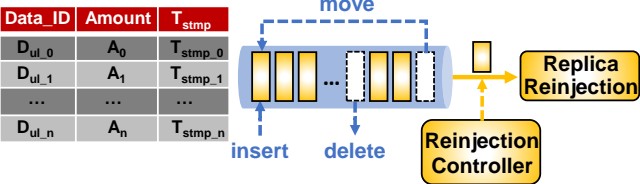

**Figure 9: Sender-side reinjection queue.**

where $\text{sigmoid}_\alpha(x) = \frac{1}{1 + e^{-\alpha \cdot x}}$. The sigmoid function reflects the introduced traffic cost. When $A_{thres} > 2$, $\text{sigmoid}_\alpha(T_{thres} - 2)$ will rapidly become larger, resulting in a worsened utility. $\alpha$ is the adjustable factor ($\alpha > 0$). By default, we set $\alpha = 0.1$. $\alpha$ reflects the importance of extra traffic cost, in which $\alpha \to 0$ shows recovery latency is the prioritized optimized target while $\alpha \to +\infty$ indicates the extra traffic cost should be seriously considered. Then AutoRec computes the optimization effects $R_\Delta$ by comparing $u_i$ to $u_{i-1}$ as follows:

$$R_\Delta = \frac{u_{i-1} - u_i}{u_{i-1}} \tag{2}$$

**Calculation of the expected waiting time.** We use $E_{waiting}$ to denote the expected time at the receiver to wait for the loss recovery. Since AutoRec is a sender-side modification, we estimate $E_{waiting}$ at the sender instead as follows:

$$E_{waiting} = (1-r) \cdot SRTT \cdot \sum_{k=0}^{n-1} (k+1) \cdot r^k \tag{3}$$

where $r$ is the actual loss rate when performing loss recoveries, and $n$ is the maximum retransmission times. The smaller the SRTT, $r$, and $n$ values are, the better the current network condition is, indicating that the receiver's expected waiting time for a single packet loss is relatively small.

**Determination of the redundancy level.** We use $DI_i$ to denote the $i^{th}$ decision interval. The Redundancy Adapter determines the redundancy level of $DI_{i+1}$ according to the redundancy level of $DI_i$. Specifically, **if** $E_{waiting}$ less than its threshold $\Theta_{thres}$, especially for the live streams with lower loss rate or shorter SRTT, we will set $A_{thres}^{i+1}$ equal to 0 (i.e., do not perform reinjection). Note that $\Theta_{thres}$ tries to reflect the time length of player cached available data. By setting a larger $\Theta_{thres}$, AutoRec can achieve targeted optimizations for those live streams that take longer to complete loss recovery. **Else**, we will calculate $A_{thres}^{i+1}$ as follows:

$$A_{thres}^{i+1} = \begin{cases} A_{thres}^i - \lambda, & R_\Delta \in (-\infty, -R_{nor}) \\ A_{thres}^i, & R_\Delta \in [-R_{nor}, R_{nor}] \\ A_{thres}^i + \lambda, & R_\Delta \in (R_{nor}, +\infty) \end{cases} \tag{4}$$

where $\lambda$ records the adjustment direction from $D_{i-1}$'s decision ($A_{thres}^{i-1}$) to $D_i$'s decision $A_{thres}^i$, where $\lambda = 1$ if $A_{thres}^i > A_{thres}^{i-1}$, and $\lambda = -1$, otherwise. $R_{nor}$ is the normal disturbance ratio that tries to filter the non-AutoRec impacts on optimization effects. For example, if $A_{thres}^i$ keeps unchangeable (compared to its previous value), the utility $u$ can also become better or worse, which is caused by the dynamic network status or congestion control.

To avoid the local optima issue incurred by this utility-powered optimization, the Redundancy Adapter enables the decision module to randomly choose a different redundancy level ($A_{thres}^{rand}$) from $[0, 4]$

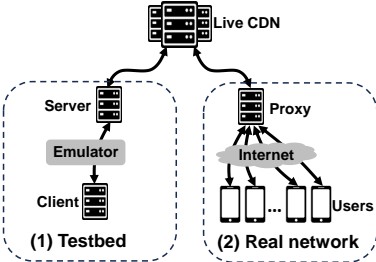

**Figure 10: Testbed and real-world deployments.**

if the redundancy levels ($A_{thres}^{old}$) selected in the past several DIs (e.g., 10) keep unchangeable. In this case, if the newly introduced $R_\Delta \leq R_{nor}$, the configured $T_{thres}$ will fall back from $T_{thres}^{rand}$ to $T_{thres}^{old}$; otherwise (i.e., $R_\Delta > R_{nor}$), the decision module will continue to determine the next $A_{thres}$ based on Eq. 4.

## 4.2 Reinjection Controller

We use $D_{ul}$ to denote the lost data that has been resent but unacknowledged by its receiver. The data $D_{ul}$ is stored and managed in a reinjection queue added at the sender. To decide when to retransmit the replicas, we introduce the Reinjection Controller which enables sending replicas of the lost packet if the stream is in off-mode. The Reinjection Controller also adopts opportunistic reinjection to further reduce the loss recovery latency while keeping the bandwidth contention for unlost data transmission within a safe limit.

**Queue management.** In AutoRec, a reinjection queue will be created by a traffic sender when a new connection of live streaming is established, which will be released once this connection is closed. AutoRec enables senders to update the reinjection queue (using the following operations) for each live stream if any lost data has been resent or acknowledged, as Figure 9 shows. The detected lost data will be inserted into the end of the reinjection queue after it has been retransmitted. The $D_{ul}$ will be deleted from reinjection queue when (i) the resent $D_{ul}$ is acknowledged by its receiver, or (ii) reinjection times $A_i$ exceeds its threshold $A_{thres}$. The $D_{ul}$ will be moved from the head to the end of the reinjection queue if it has been resent again, which is sorted by its last retransmission time. Besides, the AutoRec sender enables a status table for each reinjection queue, which records the reinjection amount ($A_i$) that has been performed, timestamp ($T_{stmp}$) of $D_{ul}$'s last (reinjected) retransmission and $D_{ul}$ identification (Data_ID)[1], as Figure 9 shows.

**Opportunistic reinjection.** To address the issue of unevenly distributed on-off mode and conduct replica reinjection more timely, AutoRec enables traffic senders to inject packets even during onmode. As illustrated in Figure 9, in the status table of the reinjection queue, $T_{stmp}$ can be leveraged to compute $D_{ul}$'s "silence" duration since its last retransmission (or reinjection). Once it exceeds the threshold $T_{thres}$, $D_{ul}$ will be fetched from the head of the reinjection queue and then resent out regardless of whether the off-mode is entered or not. Note that for achieving well-distributed loss reinjections (the challenge in §2.2), the threshold $T_{thres}$ is updated based

---

[1] In this paper, Data_ID can recognized as the packet number (pkt_num) in TCP or the stream offset (stream_offset) in QUIC.

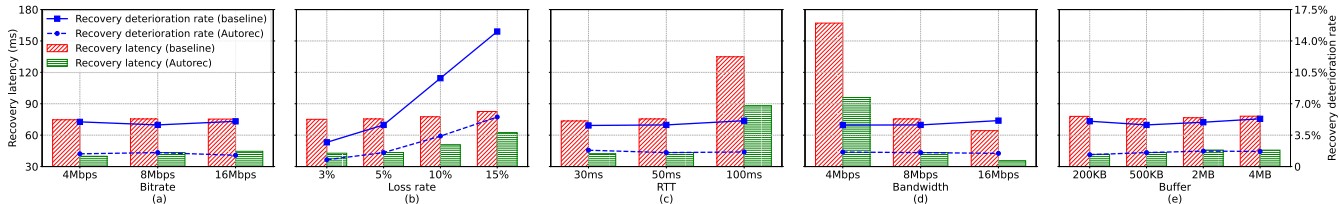

**Figure 11: recovery latency benefits in testbed experiments.**

on the latest $A_{thres}$ as follows:

$$T_{thres} = \frac{T_{unit}}{A_{thres} + 1} \tag{5}$$

It is also worth noting that the smaller the $T_{thres}$, the larger the reinjection overhead. The goodput might decrease due to the bandwidth contention from the reinjected packets. However, the Redundancy Adapter in AutoRec has already considered the impact of reinjection on goodput (see Eq. 1), this helps AutoRec maintain the incurred overhead within an acceptable range.

# 5 EXPERIMENTAL EVALUATION

We perform the experimental evaluations on our established testbed and the real networks, respectively, whose traffic senders (deploying AutoRec) can pull and transmit the requested live streaming from our live CDN to its client or real-network users, as Figure 10 shows. The AutoRec prototype is implemented based on QUIC protocol [23] (with LSQUIC Q043) [25] and NGINX architecture (with nginx 1.17.3) [26], which consists of 900+ lines of code without any client-side modification. The testbed server and client are running on CentOS Linux release 7.9 with Intel(R) Xeon(R) CPU E5-2670 v3 @ 2.30GHz (E5-2620 v3 @ 2.40GHz), 48 (24) processors, 62GB memory and 1000Mbps NIC. The employed network emulator is HoloWAN ultimate 2600u which supports 0∼1000Mbps bandwidth, 0∼10s delay, 0∼100% loss rate and 0∼1000GB buffer length. In this section, BBR (with version 1)[27] scheme is leveraged for congestion controls. The baseline scheme is the typical ARQ paradigm that will retransmit one recovery packet once a packet is detected lost. Each-stream recovery latency in this section is displayed only for the lost data, whose recovery requires at least two retransmissions.

To better evaluate AutoRec, we define a new metric called *recovery deterioration rate*, which refers to the ratio between the amount of lost data ($D_k$) that takes two or more $T_{unit}$ (i.e., recovery latency $\geq T_{unit}$) to be recovered, to the amount of all lost data. For example, there are 2 packets (one is in Figure 6(a) and the other is in Figure 6(b)) detected lost, which require 1 and 2 $T_{unit}$ for their successful recoveries (at $t_4$ and $t_7$), respectively. Then, we can learn $D_k = 1$ and recovery deterioration rate = 50%.

## 5.1 Testbed Evaluation

The testbed experiments will be performed to demonstrate the practicability of AutoRec. Unless otherwise declared, we use $R_{nor}$ = 3%, $A_{thres} \in [0, 4]$ and $\Theta_{thres}$ = 50ms. The basic environment is configured as follows: ∼8Mbps bitrate of live streams, 5% loss rate, 50ms RTT, 8Mbps bandwidth (BW) and 500KB network buffer. Each obtained metric is the average value of over 100 sets of experiments, in which each live stream will last for 60 seconds.

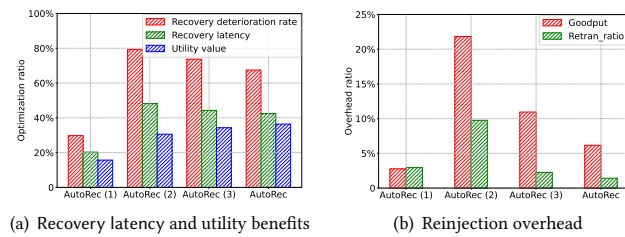

(a) Recovery latency and utility benefits     (b) Reinjection overhead

**Figure 12: The performance of specially-designed functions.**

**Benefits under different network environments.** AutoRec can introduce consistent recovery latency enhancements under variable network conditions as Figure 11 shows. We can learn the following results. (i) The changes of either video bitrate or bottleneck link buffer have fewer effects on our cared recovery latency of interest, in which recovery deterioration rate can be optimized by 69.0% from 4.94% (of baseline) to 1.53% (of AutoRec) and recovery latency will be lowered by 43.0% from 76.1ms to 43.4ms, on average. (ii) Higher loss rate can easily cause worse recovery deterioration rate that is approximately equal to the actual loss rate, which can be optimized to 0.8%, 1.6%, 3.5% and 5.7% under the loss rate values of 3%, 5%, 10% and 15%, respectively. By contrast, recovery latency reductions keep a stable range of 27ms∼32ms when loss rate≤10%. (iii) Larger-RTT live streaming actually experiences more deteriorated recovery latency, which is lowered by 34.7% from 134.9ms to 88.0ms when RTT=100ms. (iv) The bandwidth has significant effects on recovery latency, where 4Mps configuration brings recovery latency=167.1ms compared to 64.3ms of 16Mbps bandwidth.

**Performance of specially-designed functions.** To evaluate the specially-designed functions in §4, we develop three AutoRec variants, i.e., AutoRec (1), (2) and (3), to achieve (i) off-mode reinjection, (ii) off-mode and opportunistic reinjection with $A_{thres}$ = 2 and (iii) AutoRec that reinjection will perform even $E_{waiting}$ (§4.1) being too small, respectively. Figure 12 shows the benefits of recovery latency and utility as well as the recovery cost compared to the baseline. We can learn AutoRec (2) can obtain higher recovery latency benefits while incurring unacceptable overhead (e.g., 21.8% goodput deterioration). Besides, off-mode reinjection can introduce smaller reinjection costs but fail to achieve more significant recovery latency enhancements. By contrast, AutoRec with the functions of §4.1 can well balance the tradeoff between recovery latency and reinjection overhead.

**The sensitivity to pre-configured parameters.** In AutoRec, some essential parameters (i.e., $\theta_{thres}$ and $\alpha$ in §4.1) should be firstly configured to decide loss reinjection activation and evaluate last-DI's performance of AutoRec. To further explore AutoRec's sensitivity to these specific parameters, we perform sensitivity experiments on

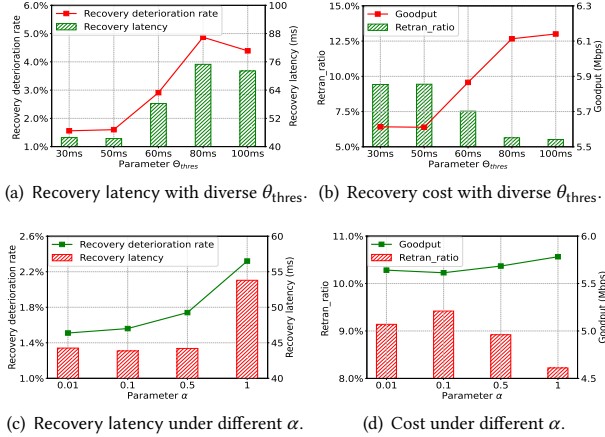

(a) Recovery latency with diverse $\theta_{\text{thres}}$. (b) Recovery cost with diverse $\theta_{\text{thres}}$.

(c) Recovery latency under different $\alpha$. (d) Cost under different $\alpha$.

**Figure 13: AutoRec sensitivity to the specific parameters.**

our testbed, as Figure 13 shows. The smaller value of either $\theta_{\text{thres}}$ or $\alpha$ can result in a more satisfied recovery latency while introducing much more reinjection cost (e.g., intolerable extra retran_ratio and deteriorated goodput). For example, $\theta_{\text{thres}}$ = 50ms can achieve significant optimization for recovery latency (= 43.4ms) and recovery deterioration rate (=1.6%), but also incurs an extra 3.9% retran_ratio and 8.6% goodput decrease, compared to $\theta_{\text{thres}}$ = 100ms. Meanwhile, recovery deterioration rate = 1.51% and recovery latency = 44.3ms are actually incurred with $\alpha$ = 0.01, compared to the corresponding 2.32% and 53.8ms of $\alpha$ = 1, respectively. However, the configuration of $\alpha$ = 0.01 also introduces an extra ~1.0% retran_ratio and the decreased goodput by 2.5%, compared to the values of $\alpha$ = 1, respectively. For $\alpha \in \{0.01, 0.1, 0.5\}$, AutoRec keeps insensitive to these parameters, in which the obtained recovery latency benefits and the caused recovery cost are close to each other.

## 5.2 Real-Network Evaluation

To further explore AutoRec's performance, we deploy AutoRec prototype in our CDN proxy and evaluate receiver-side video freezes, recovery latency benefits and reinjection cost.

**Deployment experiences.** For better deploying AutoRec in the real networks, the following rules are obeyed, which are based on our experiences for optimizing player video freezing. The specific parameter $\alpha$ of the AutoRec utility function is configured as 0.1. Meanwhile, $\theta_{\text{thres}}$ should be set as the value between the 50th- and 80th-percentile SRTT. To avoid the issues caused by higher loss rate, the deployed AutoRec is recommended to be turned off or kept inactivated if the newly monitored loss rate > 10%.

**Recovery latency benefits.** AutoRec keeps continuous optimizations for recovery latency and recovery deterioration rate in the real network, as Figure 14(a) shows. We can learn the average recovery latency and recovery deterioration rate can be lowered by the ratio of 24.6% and 37.1%, whose values are reduced from 123.2ms and 6.8% to 92.9ms and 4.3%, respectively. In particular, high-percentile (i.e., 95th-) recovery latency and maximum recovery latency can be decreased by 145ms and 384ms, which means 5% receivers take 145ms (up to 384ms) less to wait for recovering some loss if the first-time resent data is detected lost again.

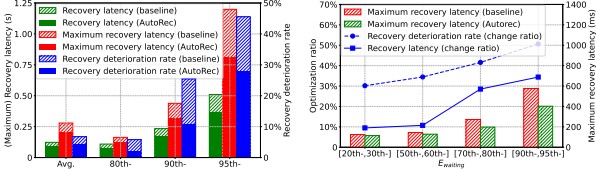

(a) recovery latency and recovery dete- (b) recovery latency vs. $\mathsf{E_{waiting}}$ ranges
rioration rate benefits

**Figure 14: AutoRec benefits in the real network.**

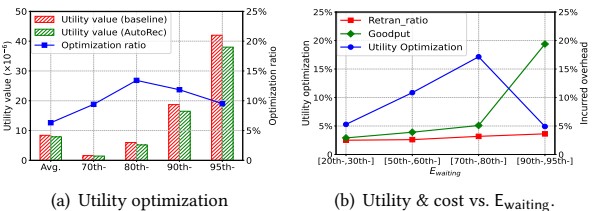

(a) Utility optimization (b) Utility & cost vs. $\mathsf{E_{waiting}}$.

**Figure 15: AutoRec overhead in the real network.**

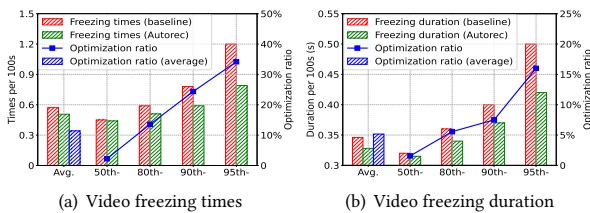

(a) Video freezing times (b) Video freezing duration

**Figure 16: The video freezing changes in the real network.**

As $\mathsf{E_{waiting}}$ represents the expected time for loss recovery from the client side (§4.1), we evaluate the recovery latency benefits under the different ranges of $\mathsf{E_{waiting}}$. Figure 14(b) shows the optimization ratios of recovery deterioration rate (30% ~ 50%) and recovery latency (10%~35%) become obvious, especially for those live streams with larger $\mathsf{E_{waiting}}$ values. In particular, 174.0ms (314.2ms) maximum recovery latency is lowered at the above ranges of $\mathsf{E_{waiting}}$. Therefore, AutoRec can achieve targeted optimization for recovery latency of large-$\mathsf{E_{waiting}}$ live streams.

**Utility and reinjection overhead.** In the real network, the deployed AutoRec enables utility optimizations without incurring non-trivial reinjection overhead. From Figure 15(a), we can learn the average utility can be optimized by 6.3%, in which the optimization ratios of 13.4% are achieved for 80th- percentile utility value, respectively. As for the incurred cost, Figure 15(b) shows the utility optimization will first become more obvious (up to ~17%) with the increased (top 80%) $\mathsf{E_{waiting}}$, which also introduces the controllable goodput deterioration of 2.5%~5.1% and retran_ratio of up to 3.6%. Besides, the tradeoff between recovery latency and recovery overhead will become difficult to balance in those streams with the last 10% $\mathsf{E_{waiting}}$, whose goodput is seriously affected by reinjection.

**Video freezing.** The AutoRec performance can be further evaluated by the observed client-side player video freezing, including its frequency and duration. As Figure 16(a) shows, the average freezing times (per 100s) can be optimized by 11.4% from 0.57 to 0.51, whose 90th- and 95th-percentile can be lowered by 24.4% and 34.1%, respectively. Besides, AutoRec also optimizes freezing

duration (per 100s) by 5.2%, in which the 95th-percentile value is reduced by ~80ms (with the ratio of 16.0%), as Figure 16(b) shows. These results demonstrate AutoRec is worthwhile for optimizing the timeliness of loss recovery and client-side video freezing.

## 6 RELATED WORK

**Adaption to on-off traffic pattern.** To the best of our knowledge, almost all prior works [28–36] fall into the category of adapting congestion control to on-off traffic patterns. For example, Zhang et al. [33] proposed a TCP variant to overcome the challenges, in which the on-off traffic pattern disturbs the increase of the TCP congestion window and triggers packet loss at the beginning of the ON period. This paper does not focus on the congestion control issues of live streaming. Instead, we take a first step toward taming the loss tolerance control under the on-off pattern for live streaming.

**Loss tolerance control for live streaming.** To achieve efficient loss recovery and optimize client-side video freezes, many studies have been proposed to enhance the data timeliness of live streaming. The key ideas of these works include injecting supplement data (e.g., FEC [5–7] and multi-path retransmissions [8, 9]), ignoring some non-critical losses (e.g., application-level controls [12–14] and semi-reliable transmissions [10, 11]). However, in commercial large-scale live-streaming product networks, the CDN vendor is mainly responsible for optimizing the end-to-end transmission performance and has control rights only on the sender side rather than the client side. Therefore, the above arts that apply client-side modification cannot meet the requirements of real-world deployment. In this paper, we propose a sender-side approach from the perspective of CDN vendors.

**Advancements upon ARQ.** Modern CDN vendors simply apply the ARQ for loss tolerance control in commercial live-streaming services. However, the legacy ARQ-based loss recovery is far from satisfactory. There also exist many studies [37–43] that improve the performance of ARQ by introducing redundancy to loss recovery. These works, however, suffer from obviously-deteriorated goodput since the inserted extra packets occupy the sender-side and in-network resources. This paper overcomes the above challenges by taking full advantage of the off-mode of live streaming.

## 7 CONCLUSION

Retransmission in itself doesn't impede slow loss recovery; rather, it's the loss of retransmission that presents the true challenge. This paper proposes AutoRec to accelerate loss recovery by allowing senders to reinject loss duplicates smartly. AutoRe's ingenuity is demonstrated by its transformation of weaknesses into strengths. It employs on-off mode switching — a feature typically challenging for transmission control — to its advantage. This approach not only facilitates quicker recovery from packet loss but also ensures that the existing data transmission on the connection remains unaffected. Both testbed evaluations and real-world deployments demonstrate the practicability and profitability of AutoRec. Currently, AutoRec has been deployed on one of the world's largest CDN vendors [reference hidden anonymously], serving thousands of millions of live-streaming users worldwide.

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
