# OpenReview forum: "Toward Timeliness-Enhanced Loss Recovery for Large-Scale Live Streaming"
_acmmm.org/ACMMM/2024/Conference — MM2024 Oral_

### Official Review · Reviewer_Cruc · 2024-05-20

**Rating:** 4
**Confidence:** 3

**Summary:**

The authors investigate how to deal with the interruptions of media being played out due to retransmissions that typically use ARQ-style error recovery.  The authors have conducted a measurement study, proposed a solution, and then evaluated the solution.

The basics of the idea are to recognize that for live streaming there are periods of time for which there is no encoded video available for transmission (called an off period).  This time can be used for preempting losses in the authors' proposed solution.

**Strengths:**

There are some interesting ideas here.  And if the results are accurate, there seems to be some benefit to the solution proposed by the authors.

**Limitations:**

The challenge with this paper is that there are some descriptions that are missing that would both improve readability, but more importantly, make the results easier to understand and digest.

For example, not a lot is said about the measurement study.  There are some fairly high-level graphs from the study, but the section leaves almost as many questions as it tries to answer.  The graphs show results for a number of countries, but they are very high level, so high level that it is hard to understand the point of the results.  One question in particular is how much of a problem is video freezing for live streaming from CDNs.  In most developed countries it seems like a pretty rare event.  The authors would do well to better motivate just how significant of a problem video freezing is.

The intuition behind the authors' solution makes sense, but there seems very little (if any) discussion of tradeoffs.  It seems the authors are essentially proposing guessing at what data might be lost or at least which lost data is likely to cause interruptions and then re-re-transmit that data.  Sending more data into the network generally doesn't help when there is congestion, but obviously not all loss is due to congestion (the assumptions TCP make, for example, are quite out-dated).  But it would be useful to better understand the tradeoffs and overhead of the solution.

Finally, the authors work is largely predicated on an assumption that no changes can be made to the client.  It isn't clear that this is an accurate assumption.  For example, decoders and the surrounding execution environment are quite flexible, to the point where forward error correct is a fairly widely deployed.  How does FEC compare to the authors' solution?

Overall, while I think there are some open issues for the paper, it might generate some discussion if I correctly understand what the authors are proposing.

**Suitability:**

2

---

### Official Review · Reviewer_7DdD · 2024-05-23

**Rating:** 4
**Confidence:** 2

**Summary:**

This paper introduces AutoRec, a mechanism to improve loss recovery in large-scale live streaming services without client-side modifications. AutoRec leverages the on-off mode switching in live streams to reduce recovery latency, transforming a disadvantage into an advantage for loss tolerance control. It employs an online learning-based policy to dynamically adjust the number of packet replicas, balancing recovery latency and overhead. The implementation of AutoRec shows improvements in reducing the times and duration of video freezing during live streaming.

**Strengths:**

1.	The topic is interesting. The paper includes a measurement study with live streams to understand loss dynamics and live streaming characteristics.

2.	The background is clearly presented. The authors provide adequate information about the topic.

3.	The authors demonstrate practicability and profitability through testbed and real-world deployments, reducing video freezing times and duration.

**Limitations:**

1.	The major flaw of this paper is that the motivation is not clear to me. It is tough to understand the limitations of existing schemes and the challenges you want to overcome from the Abstract.

2.	What are the disadvantages of on-off mode switching?

3.	In the introduction, the authors claim that the legacy ARQ-based loss recovery is far from satisfactory according to the performed large-scale measurements. But under what scenario? You have spent too much effort on introducing these quantitative results without briefly introducing some crucial settings, making it hard for readers to appreciate the significance of these quantitative results.

4.	Many important terminologies need further introduction. For example, what is on-off mode switching?

5.	In Section 2, more details about the large-scale measurements should be given. More information about the real-world logs should also be provided.

6.	Further, I needed clarification about the observation of the dynamics of loss. What is the purpose of you showing the loss rate across different regions?

7.	Many important parameters in your experiments need further discussion/justification.

**Suitability:**

3

---

### Official Review · Reviewer_ske1 · 2024-05-24

**Rating:** 4
**Confidence:** 2

**Summary:**

Proposes a way of taking advantage of off-mode live streaming to retransmit lost packets more efficiently than standard ARQ schemes: the receiver should be able to recover from the lost packet more quickly than using ARQ. Retransmissions can also be done during on-mode as long as they don't significantly impact new packet transmissions. Experimental results from a testbed and an actual network show that the proposed approach can achieve improved performance relative to a standard ARQ approach.

**Strengths:**

The idea of sending multiple retransmissions of lost packets while not disturbing new packet transmissions is innovative, and the proposed approach (AutoRec) is well explained.

The experimental results show that, at least in the conditions studied, AutoRec can improve the performance seen by live stream users.

**Limitations:**

Equation (3) computes the expected waiting time but this expression may be inaccurate e.g. requires knowing/estimating r and SRTT
 - could an observation-based approach be used instead to estimate E_waiting ?

Lots of parameters to be chosen e.g. 𝛼 , Θ_thres , R_nor
- some sensitivity results are shown for some of these parameters, but there is still a question over how they should be chosen in practice.

"To better evaluate AutoRec, we define a new metric called recovery deterioration rate"
- why not some measure or estimate of user's QoE?

How would AutoRec work for packets of different priority levels? e.g. lowest-priority packet should not be retransmitted, highest-priority packet should be retransmitted at max level, etc. This does not appear to have been considered in the proposed approach.

Instead of sending replicas at different times, what about sending them simultaneously on different routes? (spatial diversity)

**Suitability:**

3

---

### Official Review · Reviewer_zi3B · 2024-05-27

**Rating:** 4
**Confidence:** 3

**Summary:**

This paper investigates loss recovery mechanisms for large-scale live streaming systems. It leverages the insight that the typically disadvantageous on-off mode switching can be transformed into a beneficial feature to reduce loss recovery latency. Building on this observation, the paper introduces an enhanced recovery mechanism called AutoRec. AutoRec employs an online learning-based policy to adapt to the dynamic nature of packet loss, effectively balancing the trade-off between recovery latency and overhead. Experimental results from both testbed and real-world deployments in commercial services demonstrate that AutoRec significantly reduces client-side video freezing.

**Strengths:**

1. The introduction presents a new approach to addressing packet loss in live streaming services. The insight of utilizing on-off mode switching for loss tolerance control transforms a typically disadvantageous traffic pattern into a strategic advantage for enhancing loss recovery. This perspective offers a fresh solution to the problem.

2. Experiments conducted on both testbed and real-world deployments validate the practicability and effectiveness of AutoRec. These evaluations demonstrate its potential to  improve performance in commercial live streaming services.

**Limitations:**

1. A lack of comparison with existing solutions: the paper  mentions the limitations of current solutions but does not provide a detailed comparison with AutoRec on the experiment section . While the novelty of AutoRec is emphasized, a more detailed discussion of its advantages over existing ARQ-based solutions and other loss tolerance mechanisms would clarify its value proposition.

2. Weak theoretical analysis: loss recovery in live streaming has been extensively studied and can be modeled as a constrained optimization problem. However, this paper falls short in providing a precise and formal problem formulation. A well-defined problem statement is essential for setting clear objectives and constraints for the proposed solution, including specifying the parameters, metrics, and conditions under which the solution is expected to operate.

**Suitability:**

2

---

### Meta-Review · Area_Chair_Eqxp · 2024-07-02

**Recommendation:** Accept (Oral)
**Confidence:** 5

**Metareview:**

This paper has received extremely consistent ratings: all four reviewers recommend acceptance. Its overall average rating places it in the top 22% of the papers, and therefore it should be considered for acceptance.